# Diapause Regulation in Newly Invaded Environments: Termination Timing Allows Matching Novel Climatic Constraints in the Box Tree Moth, *Cydalima perspectalis* (Lepidoptera: Crambidae)

**DOI:** 10.3390/insects11090629

**Published:** 2020-09-12

**Authors:** Laura Poitou, Audrey Bras, Patrick Pineau, Philippe Lorme, Alain Roques, Jérôme Rousselet, Marie-Anne Auger-Rozenberg, Mathieu Laparie

**Affiliations:** 1INRAE, URZF, F-45075 Orléans CEDEX 2, France; laura.poitou@inrae.fr (L.P.); audrey.bras@slu.se (A.B.); patrick.pineau@inrae.fr (P.P.); philippe.lorme@inrae.fr (P.L.); alain.roques@inrae.fr (A.R.); jerome.rousselet@inrae.fr (J.R.); marie-anne.auger-rozenberg@inrae.fr (M.-A.A.-R.); 2Unit of Chemical Ecology, Department of Plant Protection Biology, Swedish University of Agricultural Sciences, SE-23053 Alnarp, Sweden

**Keywords:** invasive insect, phenology, photoperiod, plasticity, quiescence, temperature, winter diapause

## Abstract

**Simple Summary:**

The box tree moth, *Cydalima perspectalis*, is an Asian pest whose rapid invasion in Europe causes considerable economic and ecological impacts. Larvae enter a winter diapause induced by photoperiod in both native and invaded ranges, but factors that trigger the return to an active phase are still unknown. Yet, identifying them is crucial to understanding how diapause end synchronizes with the end of the winter stress encountered in Europe. We tested whether activity resumption is regulated by thermal and/or photoperiodic thresholds, two factors often involved in diapause termination, by exposing diapausing caterpillars from an invaded area to crossed treatments at the laboratory. The evolution of diapause rate was monitored over time and compared to that of nearby field sites invaded. A strong positive effect of increasing temperature was found on the rate and dynamics of diapause termination, whereas no compelling effect of photoperiod appeared. Resuming development directly when main stressors fade, not in response to indirect photoperiodic cues that could be mismatched outside native areas, likely contributes to the good match observed between diapause and the new climates encountered in the invaded range. This finding can improve phenological modelling of the overwintering generation and help better mitigate its damage.

**Abstract:**

The association between indirect environmental cues that modulate insect diapause and the actual stressors is by no means granted when a species encounters new environments. The box tree moth, *Cydalima perspectalis*, is an Asian pest whose rapid invasion in Europe causes considerable economic and ecological impacts. Larvae enter a winter diapause induced by the photoperiod in both native and invaded ranges, but factors that trigger the return to an active phase are still unknown. Yet, identifying them is crucial to understand how diapause end synchronizes with the end of the winter stress encountered in Europe. To test whether activity resumption is regulated by thermal and/or photoperiodic thresholds, or additive effects between these factors often involved in diapause termination, diapausing caterpillars from an invaded area were exposed to crossed treatments at the laboratory. The evolution of diapause rate was monitored over time and compared to that of nearby field sites invaded. A strong positive effect of increasing temperature was found on the rate and dynamics of diapause termination, whereas no compelling effect of photoperiod appeared. Resuming development directly when main stressors fade, not in response to indirect photoperiodic cues that could be mismatched outside native areas, likely contributes to the good match observed between diapause and the new climates that this pest encountered in the invaded range.

## 1. Introduction

Biological invasions are the second driving force in the loss of biodiversity worldwide [1,2], and have become a major threat to the integrity of ecosystems [3]. The rate of introductions and establishment of non-native species keeps increasing considerably, with a two-fold increase in the number of established alien insects in Europe over the past few decades [4]. Most of the insect species that have arrived during the recent decades are first-time invaders, which have never been categorized as invasive elsewhere. Moreover, a larger part of the non-native insects that have entered and established themselves in Europe since the 1990s are spreading faster than ever across the continent [5]. This is a consequence of intensifying international trade, including live plants, and tourism, which together aggravated the risks of accidental propagule transport and therefore lowered the geographic barriers to species distribution [6,7].

Multiple studies have focused on the identification of life traits that likely facilitate invasive success [8,9,10], i.e., the ability to overcome the introduction, establishment and geographic expansion stages of the invasion process [11]. A number of traits have been proposed as potentially assisting invasive success, from body size [12] to multivoltine reproduction strategies [13], or phenotypic plasticity [14]. However, evidence is lacking to firmly conclude on the generic advantage of these traits, because of contradictory results between comparative studies among taxa or ecosystems. Diapause appears as a serious candidate because this resistance form improves the survival of organisms exposed to prolonged stressors, even stressors spanning beyond those that the organisms have evolved with. Indeed, diapause suppresses metabolic processes and stops morphogenesis in response to indirect environmental cues, instead of immediate exposure to the stressors [15,16]. Diapause can therefore increase resistance to novel constraints, as long as predicting environmental cues are encountered beforehand. In the rosaceous leaf roller (*Choristoneura rosaceana*; Lepidoptera: Tortricinae), diapause increases insecticide resistance by decreasing larval exposure to the stressor, but is triggered upstream by short photoperiods and high temperatures [17]. Diapause resistance form may be key in mitigating abiotic conditions encountered by a number of alien arthropods during the transport step, or to novel abiotic and biotic conditions in the receiving environment. For instance, egg diapause was shown to improve the survival of multiple copepod species during long distance transport in ballasts, where thermal and darkness conditions differ from those in the natural environment [18]. Diapause was also found to be central in the establishment and geographic expansion of the tiger mosquito *Aedes albopictus* (Diptera: Culididae) in North America, where winters are harsher and longer than those in tropical and subtropical Asia, its native range [19].

The box tree moth, *Cydalima perspectalis* (Lepidoptera: Crambidae), is a multivoltine species originating from Asia, which rapidly invaded Europe and Asia Minor [20,21,22]. It was first recorded in 2007, where the borders of Germany, Switzerland and France intersect. A decade later, it occurred across about 30 European countries [22,23], and was observed for the first time in North America in 2018 [24]. Its accidental introduction was caused by the commercial trade of box trees between Europe and Asia, particularly from China [25], the Asian leader of live plant trade [26]. The secondary market of infested trees within Europe and the movement of plant waste are suspected to have considerably accelerated the invasive expansion of this species [27,28].

Little is known about the native distribution of *C. perspectalis*, but its hosts occur across contrasted climatic areas such as dry winter subtropical, humid subtropical, cold semi-arid, oceanic and summer continental climates [25,29,30]. Climates encountered in the invaded range are less diverse and generally temperate (i.e., oceanic, continental and Mediterranean). In the invaded range, the pest completes two to four generations a year depending on latitude [23]. In autumn, when the photoperiod decreases to a key threshold of 13.5:10.5 (L:D) found to be similar in both invaded and native ranges, pre-imaginal instars enter into diapause [20,21,23]. Diapause has been observed from second to third instars in the invaded range, while it appears to span from the second instar to as late as the pupal stage in the native range, depending on the latitude [20,23]. Diapausing larvae spend the winter sheltered into silk cocoons tied to its host leaves. This photoperiodic induction is classic in numerous insect species, since photoperiod translates seasons in a predictable and steady way over years, regardless of meteorological noise [31]. Diapause in *C. perspectalis* occurs during the autumnal generation that will be subjected to the photoperiod threshold cue and subsequent winter stress, and does not occur in previous generations or laboratory conditions under a constant summer photoperiod [32].

Diapause in this pest has been studied mainly in native populations, but the regulation of its expression and its role in invasive success under the temperate climates encountered in Europe are so far poorly understood. Additionally, most studies have focused on diapause induction, but little is known on the factors that regulate diapause termination and the resumption of the active phase in this species. Yet, the morphogenetic pause is typically locked to prevent transient returns to normal activity before several weeks or months, even when conditions are temporarily favorable, until the unlocking is triggered by specific environmental changes perceived by the organism [31]. The coupling between environmental cues regulating diapause ending and the intensity of the winter constraint in Europe is by no means trivial for a species whose evolutionary history occurred on a different continent. Therefore, identifying the factors triggering diapause termination and the return to normal activity would allow understanding the adequacy between diapause expressed in invasive individuals and the actual seasonal constraint they are subjected to in Europe. Ultimately, such progress would enable predictions of the timing of the active phase and first defoliation damage, thereby improving control methods of this major pest. In this study, we focused on the respective roles in the diapause termination of two candidate factors known to regulate diapause across many taxa: temperature and photoperiod. The relative influence of both factors was tested in the laboratory by exposing wild-sampled diapausing larvae from an invasive population in France to crossed treatments over winter and spring. The diapause rate was monitored in the samples as well as in several invaded sites in the field for validation. We hypothesize that diapausing larvae are susceptible to threshold temperatures and photoperiods that trigger diapause termination in a large proportion of individuals, possibly with additive effects between either factor.

## 2. Materials and Methods 

### 2.1. Insect Collection

Cocoons containing overwintering larvae of *Cydalima perspectalis* were hand-collected on ornamental box trees in October 2017 at Chaon (47°36′ N, 2°9′ E) in north-central France. The photoperiod threshold of diapause induction was reached before collection (photoperiod of 11:13 (L:D) during collection day), and all individuals observed had spun their cocoons, which confirms that active development was already interrupted in nature. All collected cocoons were kept for one night at room temperature (about 20 °C), before being transferred to experimental conditions mimicking winter.

### 2.2. Conditions during Winter

The day after field collections, 1440 individuals were randomly spread in 33 plastic aerated boxes (12 × 12 × 6.5 cm), and transferred to a climatic walk-in chamber at 5.0 °C ± 1.0, with a photoperiod of 8:16 (L:D), mimicking a cold constraint close to natural winter conditions in the area. All boxes were checked regularly during winter to ensure the absence of mold. Individuals were left exposed to these conditions for 16 weeks, as [23] showed that developmental success consecutive to winter diapause is maximal in this species after a minimal cold exposure of 3.5 to 4 months. This phase of the experiment is hereafter referred to as phase A (Figure 1).

### 2.3. Instar Determination

X-ray radiography was used on all individuals to control survival after winter conditions and identify diapausing larval instars without opening cocoons. To do so, larvae showing on X-ray pictures were all visually separated in three classes according to their abdomen length: small, intermediate, and large. Because abdomen length reflects not only the developmental stage of larvae but also their nutritional status, a more accurate proxy—head capsule width—was measured in a random sub-sample (10 small, 8 intermediate, 7 large) to check the reliability abdomen size classes as a criterion for instar determination in this species. Larvae from this sub-sample were extracted from their cocoon and digitized with a modular microscope (Leica DMS1000, Leica, Wetzlar, Germany, 41.83 µm/pixel accuracy) for the measurement of head capsule maximal width. Head measurements were compared with results from [23] to determine larval instars in the sub-sample, which confirmed that abdomen size can reliably discriminate diapausing larval instars in this species (see Results). Abdomen size was therefore used to constitute distinct samples of L3 and L4 individuals for further experiments (N_L3_ = 484, N_L4_ = 656).

### 2.4. Diapause Termination Conditions

Following the prolonged exposure to conditions mimicking a cold season (phase A), 20 crossed treatments of four temperatures; T5, T10, T15 and T20 (5.0, 10.0, 15.0 and 20.0 °C ± 0.4, respectively) and five photoperiods (16:8, 14:10, 12:12, 10:14, and 8:16 L:D) were tested to terminate diapause (phase B; Figure 1). The photoperiod was set using opaque white and aerated boxes (38 × 33.5 × 17 cm) combined with an internal light source (Lédis, LED 3.5 W, 300 lumens, 4500 K, Guangzhou Lampara Electronic Co. Ltd., Guandong, China). Five boxes corresponding to the five photoperiod conditions were placed in each of the four climatic chambers. Larvae were haphazardly spread into 20 culture vials for each instar, and placed in turn in the 20 photoperiod boxes (see sample sizes per instar and treatment in Table 1).

Temperature inside each plastic photoperiod box was logged every hour during the experiment (One-Wire iButton, resolution: 0.06 °C). Actual temperatures differed from the set temperatures, likely due to slight heat accumulation caused by LED lights. However, the difference among treatments remained comprised between 3.6 and 4.8 °C, with a standard deviation comprised between 0.5 and 0.7 for termination conditions (Table 2). Conditions during the prior cold period mimicking winter conditions were also slightly warmer than planned, with an average temperature of 6.0 °C ± 1.0 instead of 5.0 °C ± 1.0. The bias being similar in all Phase B treatments, they can still be equally distinguished and compared, with the exception of the T5 condition, which was supposed to be an extension of the thermal conditions during Phase A (winter, cold period), but actually was 2.6 °C warmer.

### 2.5. Activity Monitoring

Inactive (inside their cocoon), active (observed out of their cocoon and showing mobility) and dead larvae were counted five times a week when exposed to the termination phase of the experiment (Phase B). Dead larvae were removed from the photoperiod boxes to avoid any fungi proliferation. Active larvae were immediately separated from the culture vials containing diapausing ones, but kept in the same temperature × photoperiod termination conditions to test whether they were able to resume feeding and ontogeny. Fresh box tree branches were supplied ad libitum in their vial and renewed twice a week. After five weeks of exposure to Phase B conditions, all larvae still inactive were transferred for one month to more favorable rearing conditions mimicking summer, at which diapause is presumably unnecessary (20 °C, 16:8 L:D; [32]), in an attempt to force the return to activity and test the ability to complete development.

### 2.6. Monitoring the Return to an Active Phase in the Field

To validate laboratory observations, cocoon openings and larval phenology were monitored in outdoor conditions on a weekly basis. Site A (Ardon; 47°49′ N, 1°54′ E) consisted of an experimental plot with 30 young box trees of similar size, all planted simultaneously two years before the experiment and left untreated to allow natural invasion. Sites B (Huêtre; 48°10′ N, 1°47′ E) and C (Bucy-Saint-Liphard; 47°56′ N, 1°45′ E) both corresponded to sub-urban areas, where older and larger box trees were found naturally attacked by the insect. The young experimental trees in site A were checked between February and May 2018 using a standard sampling effort set to three minutes per observer per tree during all the monitoring period (i.e., 30 monitoring units per event). One monitoring event was skipped in March due to snowfalls that would have biased the sampling efficiency, and because diapause termination and locomotor activity were not expected during such a cold event. The protocol was adapted in sites B and C due to fewer but bigger trees: in each site, two ornamental box trees were checked with a sampling effort set to 15 min per observer, with each observer focusing on distinct sides of the trees (totaling six monitoring units per event in both sites B and C). Meteorological data were compiled from a climatic station located in site A, and a climatic station located within 4 and 5 km of sites B and C.

In all three sites, the weekly counting was carried out until the observed rate of diapause in either population dropped to 0. To assess larval phenology, up to 30 active larvae were collected at each monitoring event and each site, and stored in 70% ethanol until later instar determination (using head capsule measurements).

### 2.7. Statistical Analyses

Differences in head capsule width between classes were tested with a one-way ANOVA followed by a Tukey post-hoc test. To investigate the effects of temperature and photoperiod on the evolution of diapause rate over time, count data were analyzed as survival data. The statistical toolkit of survival analyses is based on the occurrence of events, i.e., an irreversible status change such as the death of an individuals. Likewise, the resumption of activity after diapause inside cocoons in the present experiment can be considered such an irreversible event. Kaplan-Meier representations were used to plot the percentage of inactive larvae over time in the different experimental conditions. In those representations, the occurrence of one or multiple events is marked by a vertical segment, followed by a horizontal segment until the next event occurrence. The effects of temperature, photoperiod and instar on the distribution of diapause rate over time in each sample were tested independently using log-rank tests. Temperature effect within both instar and photoperiod conditions was further investigated with pairwise comparisons. A Benjamini-Hochberg correction was used to deal with false discovery rate (FDR) due to consecutive tests. Finally, exposure time until 50% of individuals terminated diapause and resumed activity (hereafter referred to as T_50%_) was calculated in each condition, and the effects of temperature, photoperiod and instar were tested with a two-way ANOVA followed by a Tukey post-hoc test. All analyses were done using R version 3.5.0 [33].

## 3. Results

### 3.1. Instar Determination

Significant differences in head capsule width (ANOVA F_df_ = 63.027_2_, *p*-value < 0.0001) validated the presence of three distinct instars (L3, L4 and L5) within the subsample used for instar determination (Figure 2). Among the subsample, 100% of the individuals were correctly assigned to their validated instar when using only visual categories of abdomen size, thereby confirming the reliability of this proxy in this species. Categories of abdomen size could therefore be used to reliably sort all remaining individuals into distinct groups of L3 and L4 for the main experiment; L5 were not numerous enough (only 25) to be used as a factor, and were therefore discarded.

### 3.2. Effects of Temperature, Photoperiod and Larval Instar on Survival and Activity

Only 2% of individuals died over the 16 weeks of phase A (winter conditions, 5 °C 8:16 L:D); similarly, mortality during the five weeks of phase B did not exceed 10 and 6% in L3 and L4, respectively. Due to this high survival in all groups, no further statistical analyses were conducted.

In all statistical groups where at least 50% of individuals resumed their activity during phase B, the exposure time until this threshold was reached (T_50%_) was calculated (Figure 3). It significantly decreased with increasing temperature (ANOVA, F_df_ = 62.48_3_, *p*-value < 0.0001), but was not significantly influenced by photoperiod (ANOVA, F_df_ = 1.39_4_, *p*-value > 0.05) or instar (ANOVA, F_df_ = 0.69_1_, *p*-value > 0.05). Regardless of other factors, T_50%_ decreased linearly from T5 to T15, from about 25 days to 5 days, and then became asymptotic between T15 and T20 (Tukey post-hoc, ɑ = 0.05), with no apparent influence of instar on the shape of the decreasing trend when plotting L3 and L4 concurrently (Figure 3A). Contrariwise, while T_50%_ varied greatly among photoperiod conditions within one thermal treatment, the variations appeared erratic and unstructured (Figure 3B).

As opposed to the summary variable T_50%_, the overall evolution of inactive larvae over time was significantly altered by all three factors, however the influence of the thermal effect strongly prevailed (Table 3). Kaplan-Meïer curves depicting the decrease of inactive larvae over time (i.e., the accumulation of resumed activity within samples) confirmed steeper decreases and faster kinetics in warmer temperatures compared to cooler ones, regardless of photoperiod and instar (log-rank tests, Table 3; Figure 4). No individuals were found to resume activity and exit their cocoon during the prior cold period mimicking winter conditions (all Kaplan-Meïer curves start at 100%), which supports that changes in inactive larvae rate over time were indeed subsequent to the exposure to the new termination conditions. Yet, while samples exposed to a cool termination condition (T5) and a winter photoperiod (8:16 L:D) (i.e., identical to the prior cold period mimicking winter) showed a slower pace of inactive larvae percentage decrease than most others samples, a fraction of individuals resumed activity during the termination phase of the experiment. This may result from a slight unwanted warming of 2.6 °C, compared to the walk-in chamber where samples were kept during Phase A (Table 2). Interestingly, inactive larvae percentage decreased the least in the samples exposed to similar T5 conditions but exposed to a summer photoperiod (16:8 L:D).

Finally, when remaining diapausing larvae were transferred to 20 °C 16:8 L:D after phase B, we only observed no activity resumption in individuals originating 15 and 20 °C conditions, but the sample size was low due to most individuals, exiting cocoons during phase B (Appendix A). This contrasts with samples from 5 °C phase B, where nearly half of the transferred individuals resumed their activity. In samples from 10 °C phase B, only a small fraction of individuals resumed their activity after the transfer (Appendix A).

### 3.3. Activity and Phenology in Nature

Field monitoring on experimental trees (site A) showed a steep drop of inactive larvae percentage from 100% to 10% over a single week at the beginning of March (Figure 5). This change occurred simultaneously with a transient increase of air temperature above 10 °C (Figure 5). Inactive larvae percentage then kept decreasing down to 0% at a slower pace, together with a continuous increase of daily mean temperature. The initial decrease in inactive larvae percentage could not be observed in sites B and C, but the monitoring eventually confirmed similar percentages and trends over time during the survey. Larval phenology was found to evolve as early as the end of March in site B, but not before mid-April in the other sites, following another—more durable—increase of air temperature above 10 °C.

## 4. Discussion

Insect diapause is regulated by the combination of multiple environmental and genetic factors. Its induction and termination are usually caused by environmental cues such as photoperiod, temperature or humidity [15,34]. In the box tree moth, winter diapause is induced by a key photoperiod of 13:11 L:D [35]. [23] have shown that cold exposure during diapause improves later emergence rate. Such benefits of cold doses during diapause have been demonstrated in other Lepidoptera species showing larval diapause, such as *Choristoneura retiniana* (Tortricidae) [36] and *Pyrrharctia isabella* (Erebidae) [37] The intensity and the duration of such cold doses have long been considered prerequisites of diapause termination in insects [15,38]. However, to our knowledge, studies focusing on factors triggering diapause termination in *C. perspectalis* are scarce, despite their implications for phenology and performance in newly-colonized areas [23,39]. Identifying those factors is key to understand the synchronization between diapause regulation and the seasonal constraints that this non-native species encounters in Europe.

The exact timing of diapause termination is difficult to determine because the physiological shift to a state that unlocks ontogeny (i.e., terminated diapause) may not be immediately followed by locomotor activity or feeding if external conditions are limiting. By monitoring cocoon openings and locomotor activity, we therefore measured the actual resumption to active behaviors instead of diapause termination per se. Our study showed that both the rate and the kinetics of cocoon openings are influenced by temperature, whereas effects of photoperiod and life stage are unstructured, or concealed by the prominent impact of temperature.

### 4.1. The Influence of Temperature on Activity Resumption

The proportion of larvae exiting their cocoon and resuming their locomotor activity was maximal at T15 and T20, minimal at T5, and often intermediate at T10. This shows a positive effect of temperature on resumed activity termination that is similar to what has been reported on other Crambidae species, such as the southwestern corn borer *Diatraea grandiosella* [40]. Yet, we observed individuals exiting their cocoon even at the lowest temperature, including the control condition (5 °C, 8:16 L:D), which was a continuation of the winter conditions (phase A). Actual temperature measurements, however, showed that temperature increased by about 2.5 °C when transferring caterpillars from the winter conditions chamber to the experimental termination chamber. The latter was set to the same parameters, but the smaller air volume may have been more easily warmed by the LED spots used to manipulate photoperiod. This warming event may have been perceived as a cue by some individuals and cause the diapause termination, since not only absolute temperature, but also thermal variations are known to trigger it [15]. The decreasing number of larvae staying in their cocoons among all tested termination conditions does not allow characterizing a minimal thermal threshold. However, the pace at which most individuals resumed activity was fast at temperatures warmer than T10, and comparatively slow at T5 and T10. Moreover, the evolution over time was similar at T15 and T20, which suggests that 16.6 °C (the actual temperature measured in T15) was enough for most individuals to readily exit cocoons and resume locomotor activity. Additionally, the monitoring of remaining cocoons transferred to 20 °C 16:8 L:D after phase B also supports the fact that cool temperatures inhibited activity resumption, with the highest proportion of returns to activity in individuals originating 5 °C conditions. Together, these results suggest that the return to an active phase in the box tree moth is regulated by a combination of critical temperatures within this range and thermal variations, i.e., warming relative to winter conditions. However, discriminating the respective role of those effects in diapause termination may be challenging [15], and complex additive effects with secondary factors cannot be excluded.

For instance, photoperiod was identified as the main factor triggering diapause termination in multiple insect species, such as *Sesamia nomagrioides* (Lepidoptera: Noctuidae) [41], *Pseudopidorus fasciata* (Lepidoptera: Zygaenidae) [42], and *Pieris brassicae* (Lepidoptera: Pieridae) [43]. Yet, we found no clear effect in the box tree moth, with only a slight and unstructured influence on the general dynamics of returns to activity, while T_50%_ was not significantly impacted. [44] showed that the Asian rice borer, *Chilo suppressalis* (Lepidoptera: Crambidae), loses sensitivity to photoperiod during its winter diapause and returns to an active phase in response to warm temperatures. Progressive loss of sensitivity to photoperiod over the course of diapause has been reported in several Crambidae species, such as the Asian corn borer, *Ostrinia furnacalis*, and the beet webworm, *Loxosteges sticticalis* [44,45]. Our results are in accordance with these examples of diapause termination primarily influenced by temperature. Further studies focusing on photoperiod with larger sample sizes from different locations would be necessary to better understand the consequences of its variation during diapause, as well as whether the sensitivity of diapausing individuals decreases over time.

Our field observations of larvae exiting their cocoons as soon as daily temperatures exceeded 10 °C support the above conclusions that the resumption of activity is primarily triggered by thermal variations and thresholds in this species. Additionally, the return to activity in the invaded sites monitored appeared unrelated to photoperiod because it occurred from mid-March in sites A and B or even earlier in site C in this study whereas it was observed in April the previous years in neighboring sites (our observations) and other regions of Europe at similar latitudes (e.g., Romania; [46]). The quick response of overwintering larvae as soon as temperature increases at the end of winter raises the question of their ability to survive eventual transient colds occurring after they resumed their activity and left their cocoons, as was observed during our monitoring, with temperatures close to 0 °C.

### 4.2. Full Development in Warm Condition Only

Except in the warmest conditions where some individuals completed their development until adult emergence (seven adults, data not shown), no larvae from cooler termination conditions fed after leaving their cocoon. Instead, they remained mostly immobile, with no signs of feeding (no feces or chewing marks on supplied leaves). Likewise, the first clear phenological changes in the field occurred three weeks after the first open cocoons were observed around mid-March. The transient cold meteorological event that occurred soon after most larvae exited their cocoons probably participated in this delayed development. The congruent observations under controlled conditions and in the field together suggest that overwintering larvae are capable of terminating their diapause before optimal development conditions are met, and wait for favorable thermal windows to feed. In other words, thermal conditions that trigger the end of diapause may occur earlier and be disconnected from those required to resume active feeding and ontogeny.

### 4.3. Does the Box Tree Moth Incur a Post-Diapause Stage that May Improve Flexibility to Varying Seasonal Constraints?

Although an open cocoon unambiguously indicates that a larva is not in diapause anymore, it cannot be excluded that diapause is terminated long before the larva actually exits its cocoon. Therefore, this proxy may not reliably predict termination timing. The quick cocoon opening observed as soon as conditions warm up, together with the delayed ontogeny until temperature continues to warm up (both in nature and in controlled conditions), raise the question of the actual physiological status of larvae just before they leave their cocoon. Indeed, the diapause process is divided into several physiological phases, among which the maintenance phase, which precedes diapause termination, is associated with a locked state preventing development regardless of external favorable conditions [15]. During this phase, metabolic rate is suppressed, constant, and cannot vary until individuals are cued into the next phase called post-diapause [16]. During this final phase, individuals are only exogenously locked into a quiescent state if environmental conditions that ended maintenance (thereby triggering the termination of diapause) do not match morphogenesis thresholds. As soon as the limiting factor(s) reach the physiological thresholds, development resumes immediately [34]. This post-diapause phase can come unnoticed or appear facultative if physiological thresholds are already satisfied when maintenance is ended, because activity is then immediately visible.

Photoperiod is often the main factor regulating the maintenance phase duration [47]. This contrasts with our findings because, in all thermal conditions, some individuals returned to an active phase despite being kept in the same winter photoperiod (8:16 L:D) since they entered diapause in autumn. The cold period to which overwintering larvae are subjected to is another candidate parameter often proposed in the literature [36]. [23] showed its fitness importance in the box tree moth, with a maximal emergence success found in diapausing individuals that were exposed to more than 3.5 months of cold, and a lower emergence rate in individuals exposed to shorter cold. We used relatively mild winter conditions with 5 °C (6 °C from actual measurements), and yet observed up to 100% of larvae resuming their activity by the end of February, quickly after they were exposed to termination conditions. Consequently, the putative cold period required in this species to end the maintenance phase may be much shorter than the typical cold period; a wild individual from the population tested would be exposed to at these latitudes of the invaded range. In other words, if the cold period is a prerequisite of diapause termination in the box tree moth and plays as a switch to enable or lock the resumption of activity when temperatures temporarily increase, then this barrier appears to be lowered well before winter conditions actually end in some invaded areas. Overwintering caterpillars may be ready to leave their cocoons as soon as first favorable conditions occur, but retain the capability to wait again before resuming full activity as conditions continue to improve and stabilize.

## 5. Conclusions

The box tree moth feeds on an evergreen host. Its winter diapause is most likely a response to improve resistance to cold conditions but not a way to mitigate starvation during food depletion, as can be found in some phytophagous insects. As it is readily available for overwintering larvae that have spun cocoons onto box tree leaves, food availability probably does not constrain the timing of diapause termination. Our results show a direct relationship between the resumption of development and the actual removal of the seasonal thermal constraint diapause mitigates, instead of an indirect cue. This may facilitate the good match between overwintering larvae and the wide variety of winter harshness and duration that they can encounter across different latitudes in native and invaded regions. Our observations suggest that the maintenance phase which locks morphogenesis independently of the stress diapause allows avoiding, probably ends before the actual stress fades out in the invaded range. This enables resuming development whenever conditions become favorable, which may prove to be a significant advantage to performance in a wide range of climates despite annual and geographical variations for this species whose cycle does not depend on the phenology of its host. Further studies will be needed to firmly confirm this hypothesis, namely by targeting the transition into a post-diapause quiescent phase and the parameters that control it.

We also observed multiple larval instars undergoing diapause and successfully resuming their development after being able to pass the winter. Similar observations were reported for invasive populations in other parts of Europe but, to date, only two larval instars had been mentioned in nature [21,48], and three for experimentally induced diapause [49]. We found a greater plasticity, with at least L2 (only observed in monitored sites), L3, L4 and L5 (only in the laboratory sample) instars all being able to incur and survive winter diapause in the population we studied. It is not known whether the survival rate to diapause is similar in all stages, and whether they can equally sustain long periods of inactivity, despite different amounts of energy reserves. However, such plasticity in the ability to enter diapause at multiple stages may contribute to winter survival and later population dynamics in invaded areas. Indeed, the phenological variation typically observed at the end of summer because of overlapping generations or microclimatic differences is not necessarily filtered down by the inability of some stages to respond to the key induction photoperiod, but instead individuals at different stages of their larval life can suspend their development to resume it after winter. Such flexibility in the ontogenic requirements to enter diapause enables higher phenological variation within populations. Given the different vulnerabilities to stressors often found among larval instars, this likely translates into increased population survival to stochastic environmental stressors and, consequently, increased establishment success into novel areas. Plasticity in the ability to express diapause at multiple phenological stages and early post-diapause quiescence likely are important adaptive traits facilitating initial survival of populations when novel conditions are encountered. Adaptive plasticity in diapause regulation is therefore a good candidate that may contribute to the rapid invasive success of *C. perspectalis* in a wide range of newly colonized environments and climates.

## Figures and Tables

**Figure 1 insects-11-00629-f001:**
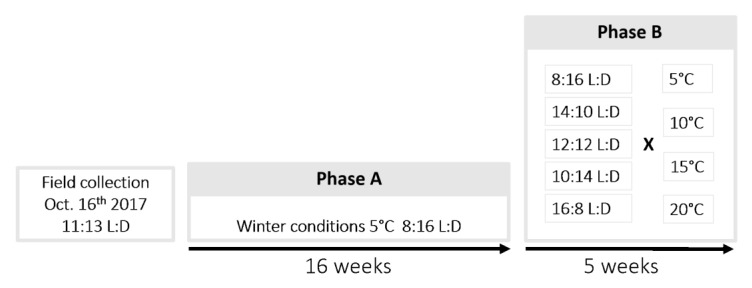
Experimental design. Phase A: Larvae collected in their cocoons were exposed to temperature and photoperiod conditions mimicking winter. Phase B: Cocoons containing larvae were split and exposed to diapause termination conditions corresponding to 20 crossed treatments of four temperatures and five photoperiods.

**Figure 2 insects-11-00629-f002:**
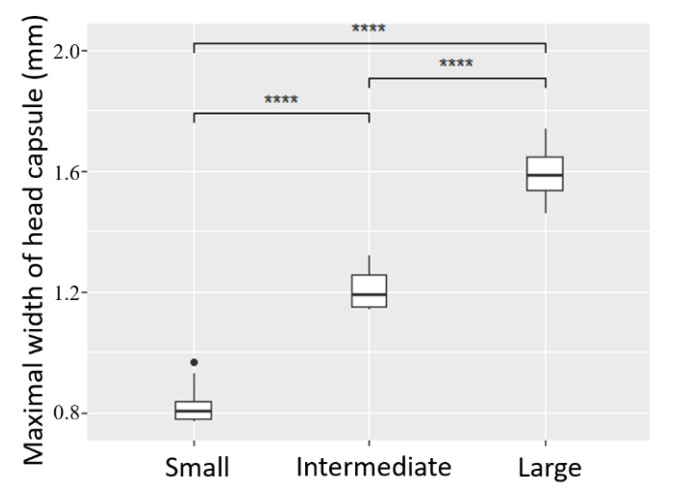
Head capsule width of a sub-sample of larvae according to abdomen length classes visually defined from X-ray pictures: Large (*n* = 7), Intermediate (*n* = 8), Small (*n* = 10). ANOVA and Tukey post-hoc test, ****: *p*-value < 0.0001.

**Figure 3 insects-11-00629-f003:**
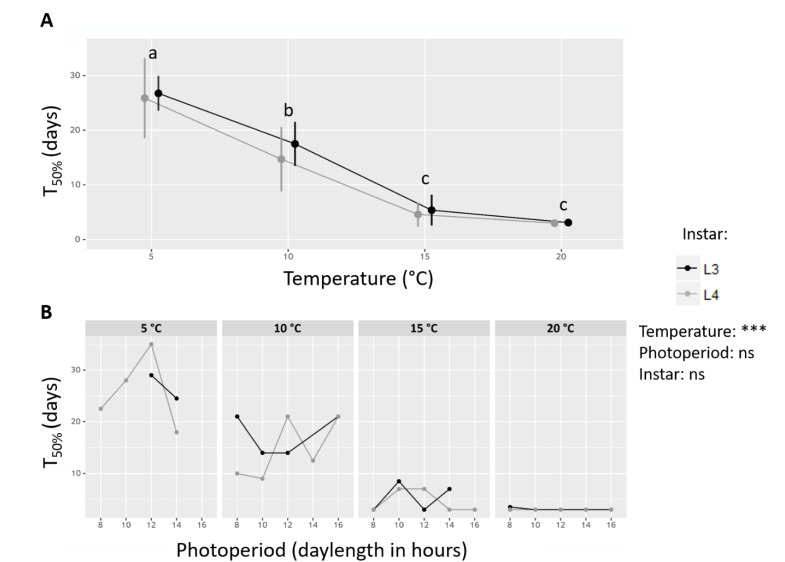
Exposure time to experimental termination conditions until 50% of larvae resumed their activity (T_50%_). (**A**) T_50%_ depending on Temperature and Instar. All photoperiod conditions are grouped. Letters indicates significant differences after a Tukey post-hoc test with a threshold of α = 0.05. Curves are shifted along the x axis to ease visualization. Error bars: standard deviation. (**B**) T_50%_ depending on Temperature, Instar and Photoperiod (only in groups where diapause rate decreased below 50% during phase B). Results of a three-way ANOVA testing the effects of Temperature, Instar and Photoperiod on T_50%_ are indicated, ns: not significant, ***: *p*-value < 0.001.

**Figure 4 insects-11-00629-f004:**
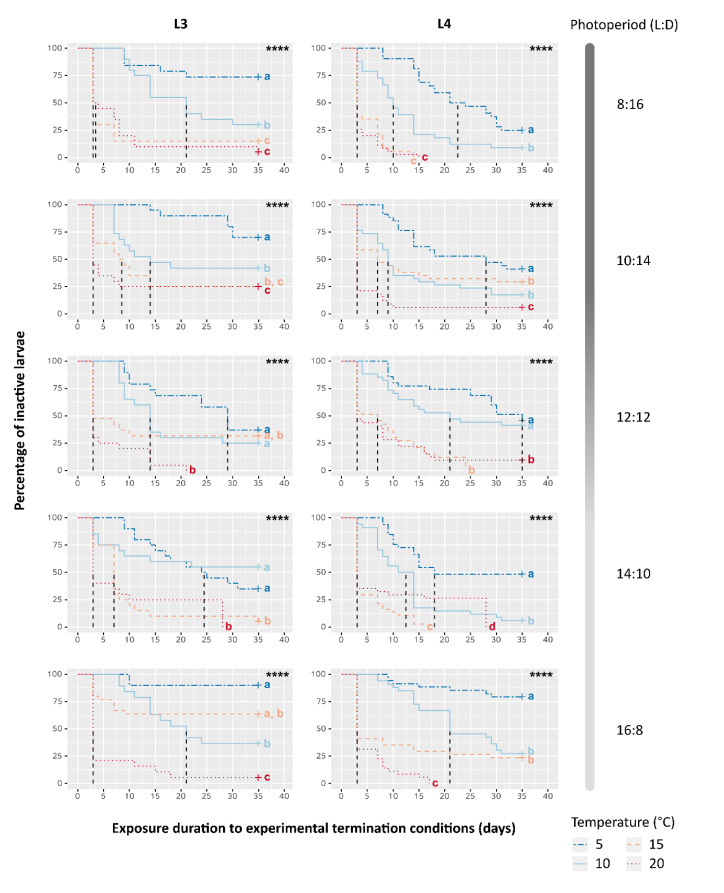
Kaplan-Meier representation of the evolution over time of the percentage of inactive larvae depending on Temperature, Photoperiod and larval instar (L3 or L4). Dashed vertical lines correspond to the exposure time until 50% of larvae resumed their activity (T_50%_). Moreover, “+” indicates data censure, i.e., the presence of larvae still inactive at the end of experiment. Log-rank test results are reported in each graph, ****: *p*-value < 0.0001. Different letters show significant differences (α = 0.05) among curves of the same graph after pairwise comparisons and a Benjamini-Hochberg correction.

**Figure 5 insects-11-00629-f005:**
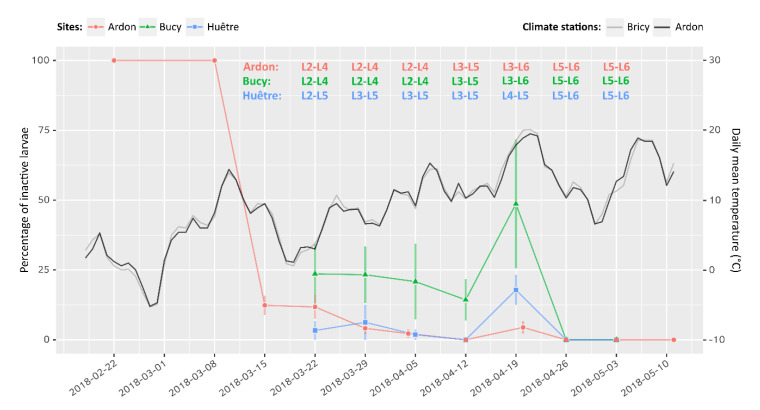
Evolution of the percentage of inactive larvae over time (left axis) and daily mean temperature (right axis) in three different locations. The Bricy climate station is located close to the Bucy–Saint–Liphard (site B) and Huêtre (site C) sites where larvae were monitored (5.2 and 4.3 km, respectively). Error bars: standard deviation. For each sampling, active larval instars were identified using head capsule width measurements on a sub-sample of 30 individuals per site. Sources of climate data: Météo France (Bricy) and INRAE (Ardon).

**Table 1 insects-11-00629-t001:** Number of larvae of third (L3) and fourth (L4) instar exposed to different thermal and photoperiod conditions during the diapause termination experiment (phase B).

Temperature (°C)	5		10		15		20	
Larval Instar	L3	L4	L3	L4	L3	L4	L3	L4
Photoperiod (L:D)								
8:16	20	34	20	34	20	34	20	34
10:14	20	34	20	34	20	34	20	34
12:12	20	34	20	34	20	34	20	34
14:10	20	34	20	34	20	34	20	34
16:8	20	34	20	34	20	34	20	34

**Table 2 insects-11-00629-t002:** Mean temperature recorded during the experiment in the different experimental conditions (phase A: Winter; and B: T5, T10, T15, and T20).

Experimental Condition	Set Temperature (°C)	Actual Mean Temperature ± SD (°C)
Winter	5	6.0 ± 1.0
Temperature 5 (T5)	5	8.6 ± 0.6
Temperature 10 (T10)	10	12.2 ± 0.5
Temperature 15 (T15)	15	16.6 ± 0.5
Temperature 20 (T20)	20	21.4 ± 0.7

**Table 3 insects-11-00629-t003:** Results after Benjamini-Hochberg correction of the log-rank tests for the three experimental factors. ****: *p*-value < 0.0001, ***: *p*-value < 0.001.

Factor	χ²	Df	*p*-Value
Temperature	351.0	3	***
Larval instar	21.1	1	****
Photoperiod	39.2	4	****

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
