# Peer review of "Diapause Regulation in Newly Invaded Environments: Termination Timing Allows Matching Novel Climatic Constraints in the Box Tree Moth, Cydalima perspectalis (Lepidoptera: Crambidae)"

_insects, 2020, doi:10.3390/insects11090629_

Round 1

Reviewer 1 Report

In this MS, Poitou et al. examined the effects of temperature and photoperiod as environmental cues of diapause termination in Cydalima perspectali which is an invasive species in Europe. Little is known about an environmental cue for diapause termination in this species and the authors tested two factors in experimental conditions and collected data under natural conditions. The results are valuable for elucidating the adaptation of invasive insects to different climate conditions. I would recommend it for acceptance after revising the a few minor points listed below are addressed.

  1. L193-196: the authors said they transferred inactive larvae to summer conditions but they didn’t show any data in Result part. If the authors can add the data, it would be helpful for readers.
  2. In some experimental groups a lot of larvae did not terminate diapause and the authors couldn’t calculate T50%. The readers cannot understand why some data points are missing in Fig 3 without seeing Fig 4. I recommend to revering the order of showing Fig3 and 4.

Author Response

We are grateful for the positive review and helpful comments we received to improve our manuscript. It was amended in the light of your comments and we believe the corrections have strengthened the manuscript. We hope that this revised version will solve the minor issues raised.

1. “L193-196: the authors said they transferred inactive larvae to summer conditions but they didn’t show any data in Result part. If the authors can add the data, it would be helpful for readers.”

We added a paragraph in the results to elaborate on this data (L286-291) as well as a supplementary table detailing the number and percentage of individuals transferred to summer conditions and observed resuming their activity. These results are discussed (L355-358).

2. “In some experimental groups a lot of larvae did not terminate diapause and the authors couldn’t calculate T50%. The readers cannot understand why some data points are missing in Fig 3 without seeing Fig 4. I recommend to revering the order of showing Fig3 and 4.”

We agree with referee 2 that the missing data points in Fig 3 were confusing. Swapping figures 3 and 4 would however imply restructuring the text to go from the most detailed results to the synthetic variables, which we believe would make the MS more difficult to read. Therefore, we proposed mentioning before Fig 3 (L250-L251) as well as in its legend (L265) that the 50% diapause threshold used to calculate T50% times plotted in Fig 3 was not reached in all conditions. The additions reads as follows:

L250-251: "In all statistical groups where at least 50% of individuals resumed their activity during phase B, the exposure time until this threshold was reached (T50%) has been calculated (Fig. 3)."

L265: "(only in groups where diapause rate decreased below 50% during phase B)”

We believe this cleared the ambiguity and improved the manuscript while keeping its current structure.

Reviewer 2 Report

Assessment of the paper entitled “Diapause regulation in newly invaded environments: termination timing allows matching novel climatic constraints in the box tree moth, Cydalima perspectalis (Lepidoptera: Crambidae)” for Insects (insects-921093)

Main comments

In this paper, the authors aimed to evaluate whether activity resumption of Cydalima perspectalis (Lepidoptera) is regulated by thermal and/or photoperiodic thresholds, two factors involved in diapause termination, by exposing diapausing caterpillars from an invaded area in Europe (France) to crossed treatments at the laboratory (five photoperiod and four temperatures). They also observed caterpillars in the field. This study is well designed, presents good results and insights on the establishment success into novel areas of an insect of economic importance. I congratulate the authors for a good piece of work. I have only minor issues (editing and style) that need to be addressed before publication.

Minor issues

Line 16, 32. Change ‘cause’ to ‘causes’

Line 99-100. Check for duplicated spaces here (after “during” and “photoperiod”)

Line 141. Change to “Fig. 1”

Line 235. Missing degrees of freedom

Line 237. Change to “Fig. 2”

Line 248, 249. I am not sure this is the best way to present test values and degrees of freedom.

Line 264, 271. ‘i.e.’ italicized?

Author Response

We are grateful for the positive review and helpful comments we received to improve our manuscript. It was amended in the light of your comments and we believe the corrections have strengthened the manuscript. We hope that this revised version will solve the minor issues raised.

1. Line 16, 32. Change ‘cause’ to ‘causes’
2. Line 99-100. Check for duplicated spaces here (after “during” and “photoperiod”)
3. Line 141. Change to “Fig. 1”
4. Line 235. Missing degrees of freedom
5. Line 237. Change to “Fig. 2”
6. Line 264, 271. ‘i.e.’ italicized?

We corrected the manuscript according to the above comments.

7. Line 248, 249. I am not sure this is the best way to present test values and degrees of freedom.

Insects guidelines did not suggest a preferred format, therefore we followed conventions seen in other journals, but we are happy to edit according to recommendations from the editor during the final editorial phases.